# An Atypical Etiology of Fungal Keratitis Caused by *Roussoella neopustulans*

**DOI:** 10.3390/jof8050507

**Published:** 2022-05-15

**Authors:** Morgana F. Voidaleski, Flavio Queiroz-Telles, Hugo T. Itikawa, Guilherme G. Müller, Bruna J. F. S. Lima, Lucas E. Trevisoli, Regielly C. R. Cognialli, Roberta C. L. Crispim, Vania A. Vicente

**Affiliations:** 1Postgraduate Program in Microbiology, Parasitology and Pathology, Biological Sciences, Department of Basic Pathology, Federal University of Parana, Curitiba 81531-980, Brazil; morganavoidaleski@gmail.com (M.F.V.); jacomel.bruna@gmail.com (B.J.F.S.L.); 2Department of Public Health, Hospital de Clínicas, Federal University of Paraná, Curitiba 80060-900, Brazil; regielly.cognialli@gmail.com; 3Hospital de Olhos Noroeste do Paraná, Cianorte 87200-000, Brazil; Itikawahugo@outlook.com; 4Faculty of Medicine, Federal University of Paraná, Curitiba 80060-000, Brazil; 5Médicos de Olhos S/A, Curitiba 80420-100, Brazil; gubertmuller@gmail.com; 6Mackenzie Evangelical School of Medicine Parana, Curitiba 80730-000, Brazil; 7Unimed Laboratório Curitiba, Curitiba 80240-030, Brazil; letrevisoli@gmail.com (L.E.T.); roberta.crispim@unimedlab.com.br (R.C.L.C.); 8Postgraduate Program in Internal Medicine and Health Science, Federal University of Parana, Curitiba 80060-900, Brazil

**Keywords:** eye infections, keratitis, *Roussoella*, natamycin

## Abstract

Fungal keratitis is caused by a wide spectrum of fungal genera, including molds and yeasts. We report a 42-year-old patient with mycotic keratitis after a direct trauma by a wood fragment. The fungal isolate was identified as *Roussoella neopustulans* by molecular methods. The treatment with topic natamycin showed progressive improvement of the visual manifestations, and following three months of therapy, the patient regained sight. We report the first case of keratitis associated with *R. neopustulans*.

## 1. Introduction

Human corneal infections are caused by bacteria, fungi, virus and protozoa, and they are globally distributed, being one of the main causes of visual health impairment and blindness in underdeveloped countries. The global incidence of fungal keratitis is around 1 to 1.4 million cases per year, with 100,000 cases evolving to eye loss and 600,000 to blindness [1]. The incidence of keratitis is difficult to characterize; studies show that it is variable worldwide. These infections are more frequent in developing countries like China and India, than in developed countries, like the United States and Australia. In the United States, the incidence of keratitis is 11 cases per 100,000 inhabitants, in Nepal, it is higher, reaching 799 cases per 100.00 inhabitants [2,3,4]. In India, a study reported over a period of 10 years and 5 months, 1360 individuals affected by ocular mycosis, confirmed by culture of keratitis samples. Another study in northern China characterized 654 patients with the same condition over a 6-year period. In comparison, Melbourne (Australia) over an 8-year period and New York in 16 years, documented mycotic keratitis in 56 and 57 patients, respectively [3].

Rapid and early diagnosis is essential to provide efficient treatment and to avoid complications, of intraocular involvement [2,4,5]. *Staphylococcus aureus*, *Pseudomonas aeruginosa*, *Streptococcus pneumoniae* and *Serratia* spp. have been commonly described as bacterial agents of infectious keratitis [3,4,5]. Among the fungi agents, *Fusarium*, *Aspergillus*, *Curvularia*, *Paecilomyces*, *Scedosporium*, *Candida* and melanized fungi such as species of *Alternaria* and *Cladosporium*, are frequently isolated. In addition, *Acanthamoeba* is most prevalent protozoa, involved in ocular infections [3,4,5].

The most relevant risk factors associated to fungal keratitis caused by filamentous fungi, are the chronic use of topical ocular corticosteroids, vegetal related trauma and the usage of contact lenses. Corneal infections caused by yeast-like fungi are more common in immunocompromised patients, or in those undergoing eye surgeries and also in contact lens usuaries [4,6,7].

According to Hofling—Lima et al. (2005) [6], fungal keratitis affects two times more men than women and is strongly associated with occupational risk. In addition, it is well known that climate conditions, host precedence (rural or urban) and profession contributes to variations on the infection process, and it is important to correlate it with frequency of case [6]. Ibrahim et al. (2012) [8] studied the seasonality of keratitis in Brazil through the evaluation of antifungal eye drop selling (AEDS) during the year. The authors demonstrated an increase of AEDS in the third trimester, when agricultural activity is more intense and the air humidity decreases [2,3].

The main clinical manifestations of infection keratitis are inflammation, redness, pain, lacrimation and blurred vision. These could vary according to the etiological agent, fungal burden, virulence factors and the efficacy of the host defense mechanisms. The therapy management shall focus on the pathogen elimination and in the minimization of inflammatory response [4,5].

Corneal smears and culture, the current gold standard mode of diagnosis, demonstrate an increasing trend in the fungal keratitis and a wide variety of fungal pathogens (>100 species) [1] including new and rare species. Mochizuki et al. (2017) [9] related the first case of *Roussoella solani* as an etiologic agent involved in keratomycosis in a human host from Japan. Species belong to *Roussoella* genus are mainly described as saprobes, e.g., *R. nitidula*, *R. intermedia*, *R. japanensis* and *R. neopustulans* isolated from decaying bamboo culm [10]. Few cases reported subcutaneous mycoses involved *R. percutanea* [11,12,13,14].

In summary, the etiology of keratitis is variable, but remarkably is the microbiologic identification of the etiologic agent as well the determination of its susceptibility profile. Diseases caused by molds causing keratitis remain under-reported due to phenotypical identification, thus molecular markers contribute to the elucidation of the etiologic agent and epidemiology of keratitis, especially in rare cases. Therefore, the aim of this study is to report the first case of fungal keratitis caused by *Roussoella neopustulans*.

## 2. Case Presentation

A 42-year-old man with no history of use of contact lenses or the presence of other ocular or systemic comorbidities, proceeding from a rural area of Parana state in south Brazil was presented to his ophthalmologist complaining of visual blurring, foreign body sensation and ocular pain in his left eye (day −7). He reported direct trauma by a wood fragment in this same eye about 10 days before (day −17). The patient used ciprofloxacin with dexamethasone drops on his own for the first few days, but ceased as symptoms worsened.

On physical examination, he referred a 20/20 visual acuity in the right eye and 20/50 in the left eye. In the examination of the anterior segment of the eye, he presented with an ulcerated lesion in the cornea with a dense whitish superficial infiltrate of 1.5 × 2.0 mm in size, with well-defined margins, located on the nasal margin of the pupil of the left eye (Figure 1B,D,E). The fundus examination showed normal macula and optic nerve in both eyes. Based on the clinical findings and the morphological features of the fungal organisms, a presumptive diagnosis of keratitis was formulated. The patient was started on Gatifloxacin 0.3% (Zymar^®^, Allergan, SP, Brazil) drops every 2 h. Despite the measures, there was no improvement in symptoms or changes in the physical exam upon reevaluation.

A corneal scraping was performed that showed septate hyphae on the Gram stain microscopic examination (Figure 1C). The patient was started on natamycin 5% (EyePharma, São Paulo-SP) eye drops every hour (day 0) and Gatifloxacin 0.3% eye drops were maintained every 6 h until growth of the fungus in culture. The patient was reevaluated every 3 to 5 days with progressive improvement of signs and symptoms.

Mycological culture was performed on Sabouraud agar (SDA) for 10 days at 30 °C. After 7 days, the culture showed a whitish filamentous cottony fungus that had no conidiogenous structures to aid morphological identification. For the molecular identification colonies cultivated on Sabouraud glucose agar (SGA) was used to DNA extraction (Figure 1A). The fungi mycelium was transferred to a 1.5 mL tube containing 100 mg of a silica:celite mixture (2:1, *w*/*w*) and 300 µL CTAB buffer [CTAB 2 % (*w*/*v*), NaCl 1.4 M, Tris-HCl 100 mM, pH 8.0; EDTA 20 mM]. Fungi cells were disrupted with a pestle for 5 min, and incubated for 10 min at 65 °C. Then, 500 µL 24:1 chloroform: isoamyl alcohol (CIA) was added and the solution was centrifuged for 10 min at 20,500× *g*. The supernatant was collected to a new tube with 2 vols of ice-cold 96% ethanol. DNA was precipitate for 2 h at −20 °C and then centrifuged for 10 min at 20,500× *g*, washed with cold 70% ethanol, drying at room temperature, and resuspended in 100 µL in ultrapure water.

Amplification of the rDNA Internal Transcribed Spacer (ITS) was performed using the ITS1 and ITS4 [15] LROR and LR5 [16] for LSU. PCR reaction mixtures were made with a total volume of 12.5 μL (1× PCR buffer, 2.0 mM MgCl_2_, 25 μM deoxynucleotide triphosphates (dNTPs)), 0.5 μM of each forward and reverse primer, 1 U of Taq DNA polymerase (Ludwig Biotec, Bela Vista, Brazil), and 10 ng of genomic DNA. PCR parameters for amplification were 95 °C for 4 min, followed by 35 cycles consisting of 95 °C for 45 s, 52 °C for 30 s, and 72 °C for 2 min, and a delay at 72 °C for 7 min, performed in an ABI Prism 2720 thermocycler (Applied Biosystems, Foster City, CA, USA). Amplicons were sequenced with BigDye Terminator cycle sequencing kit v. 3.1 (Applied Biosystems, Foster City, CA, USA) according to the manufacturer’s instructions, using the same primers of the PCR, and the amplification condition as follows: 95 °C for 1 min, followed by 30 cycles consisting of 95 °C for 10 s, 50 °C for 5 s and 60 °C.

Consensus sequence of the ITS and LSU regions was visually inspected using MEGA v.7 software [17] and compared to GenBank Blast (NCBI). The sequences of ITS were aligned with reference strains using the online MAFFT interface and the substitution model was selected for each genus using the MEGA software. Phylogeny was constructed based on ITS gene. Bayesian inference was performed in MrBayes version 3.2.1 [18] using the number of generations needed to reach split frequencies of ≤0.01. Resulting trees were plotted in FigTree v.1.4.2 [19]. The clinical strain clustered with *Roussoella neopustulans* (Figure 2). The GenBank accession numbers of the large subunit of the nuclear rRNA gene and the ITS region are OL799163 and OL799109, respectively.

After 7 days of treatment (day +7), the natamycin drops were tapered to one drop every other hour for 3 additional days, and then to one drop every 3 h for an additional 4 days, when complete healing of the lesion was observed (day +14). Finally, natamycin eye drops every 6 h were maintained for 7 extra days (day +21).

Following 3 months, the patient was asymptomatic and referred a visual acuity of 20/20 in the left eye (day +122). The examination of the anterior segment showed a superficial nummular scar that did not compromise the visual axis (Figure 1F).

## 3. Discussion

The Roussoellaceae was described by Liu et al., 2014 [10], as a new family in Pleosporales order in the class Dothideomycetes. Ahmed et al. (2014) [11] described the first report of the genus *Roussoella*, as an opportunistic pathogen causing subcutaneous mycoses on foot and ankle and the agent was identified as *Roussoella percutanea*. Afterwards, there are three reports of human infection related described in the literature. The first one is a case of phaeohyphomycosis in the ankle [12]; the second a subcutaneous mycotic cyst [13]; the third case of bursitis in a patient with osteoarthritis in the left knee joint [14]. In all of cases, the patients had a history of a renal transplant, and no trauma has been related (Table 1).

In our case, we noticed an ocular infection caused by *R. neopustulans* associated a traumatic inoculation. Mochizuki et al. (2017) [9] reported the first keratomycosis by *R. solani* in Japan, and as the other cases there was no history of ocular trauma. The keratitis is prevalent in rural workers and the ocular trauma with plant material are considered main risk factor [4,6,7] like this case that the patient reported direct trauma by a wood fragment in the affected eye.

In a prevalence study conducted by Ghosh et al. (2016) [20], *Aspergillus* spp. was the most frequent etiologic agent of fungal keratitis (47.6% of 393 isolate-confirmed cases), followed by melanized fungi (21.9%), and *Fusarium* spp. (16%). Additionally, melanized fungi showed statistical correlation with delayed diagnosis when compared to other fungi, probably because infection advances at a slower pace.

Furthermore, *R. percutanea* is the major agent of the genus and *R. solani* are the first species associated with keratomycosis. This is the first of keratitis case caused by *R. neopustulans* and the first reported of human infection associated to this species. The *Roussoella neopustulans* was firstly described by Liu et al., 2014 [12], as a saprobic organism isolated on decaying bamboo culms morphologically similar and phylogenetically related to *R. pustulans*. According to our molecular analysis, the ocular isolated belong to Roussoellaceae family close to the type strain *R. neopustulans* MFLUCC 11−0609 (Figure 2).

To date, *Roussoella* spp. was not reported in the etiology of human or zoonotic infections. The most prevalent agents causing human keratitis are *Fusarium* species such as *F. solani*, *F. verticillioides*, *F. oxysporum* and *F. dimerum* [21]. In addition, the genera *Candida*, *Paecilomyces*, *Penicillium* and *Aspergillus* have been reported as causal agents [8,21,22]. Melanized fungi such as species of *Alternaria* and *Cladosporium* are isolated as etiological agents of fungal keratitis [23,24,25].

Cutaneous *Roussoella* infections reported in the literature have been treated with voriconazole for a few weeks or months (6 weeks to 9 months) [13,15,16,17]. Ahamed et al., 2014 [11], suggest that azoles showed the highest activity against isolate *R. percutanea* than echinocandins and flucytosine, but indicate the testing of patient isolate available to the treatment. However, especially for superficial ocular fungal infections, 5% natamycin is recommended [26,27], and in this case it was the efficient therapy to progressive improvement of signs and symptoms of *R. neopustulans* keratitis. Natamycin is an antifungal used only in the treatment of fungal keratitis, being the only antifungal agent approved by the US Food and Drug Administration, with a broad spectrum of action, especially against filamentous fungi [28,29]. According to Prajna et al., 2010 [27], natamycin and voriconazole are the same to visual acuity improvement. Isavuconazole by systemic administration was successfully used by Cultrera et al. (2021) [30] to treat a case of keratitis by *Subramaniula asteroids*; a filamentous fungus. Although azole compounds have shown promise in the therapy for fungal keratitis, natamycin carries on as the therapy of choice for filamentous fungal keratitis [26,27,28,29].

In summary, the current case showed the importance of microbiologic diagnosis of human keratitis, using phenotypical and molecular methods, avoiding misdiagnosis with environmental colonizer fungi. Moreover, the molecular identification in this case was substantial to improve and confirm the conventional identification approaches. The *R. neopustulans*, as a new etiologic agent of fungal keratitis, increases the spectrum of species associated with this infection that should be integrated in the medical mycology diagnosis.

## Figures and Tables

**Figure 1 jof-08-00507-f001:**
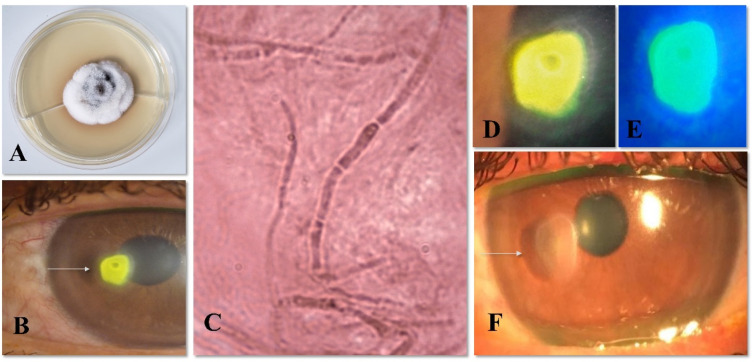
Fungal Infection keratitis clinical signs and causal agent identification (**A**): *Roussoella neopustulans* macro morphology of the colony; (**C**): direct exam showing septa hyphae by Gram staining (1000× magnification); (**B**,**D**,**E**): Nasally located fluorescein-stained corneal ulcer on presentation; (**F**): Superficial nummular scar 3 months after treatment.

**Figure 2 jof-08-00507-f002:**
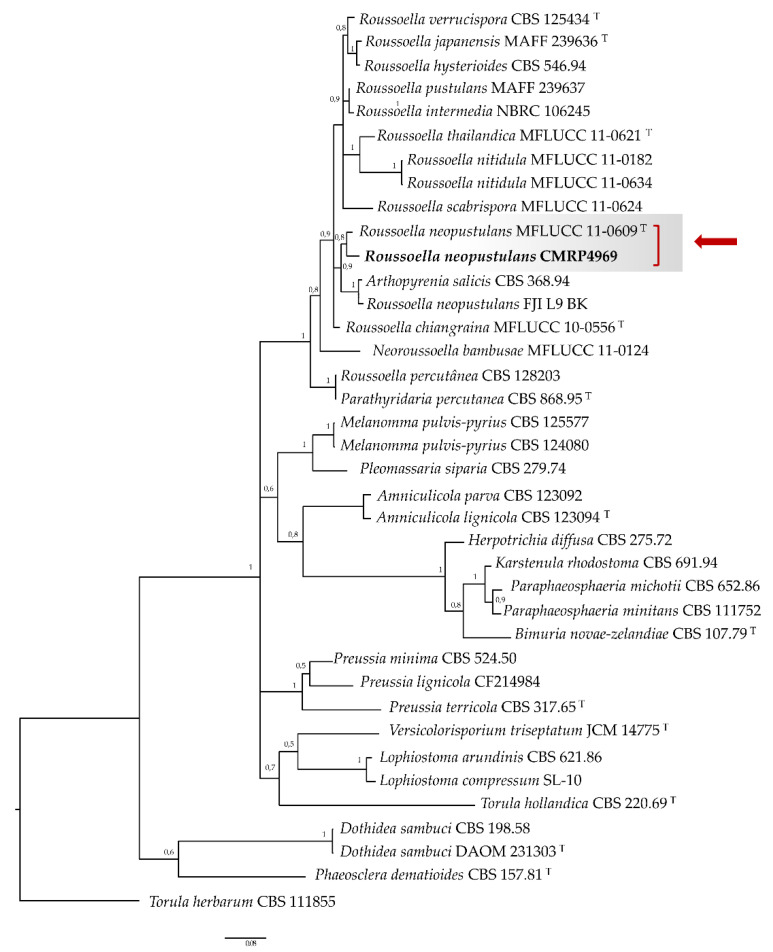
Phylogenetic tree of *Roussoella* species based on aligned rDNA Internal Transcribed Spacer (ITS) sequences, obtained by Bayesian analysis. *Torula herbarum* CBS 111855 was taken as an outgroup. T = Type strain. The sequences of this study are indicated in bold. Values of ≥95% for Bayesian probability resampled datasets are shown with the branches.

**Table 1 jof-08-00507-t001:** Clinical cases overview of *Roussoella* genus.

N.	Age	Gender	Etiologic Agent	Manifestation	Trauma	Treatment	Region	Reference
1	45	M	*Roussoella percutanea*	Subcutaneous mycosis	No related	PSCZ, VRCZ, ITCZ	USA	[11]
2	65	M	*Roussoella percutanea*	Phaeohyphomycoses	No informed	PSCZ	France	[12]
3	55	M	*Roussoella percutanea*	Bursitis	No related	VRCZ	Somalia	[14]
4	47	M	*Roussoella percutanea*	Subcutaneous mycosis	No related	VRCZ	United Kindown	[13]
5	82	M	*Roussoella solani*	Keratomycosis	No related	VRCZ, ITCZ, MCFG	Japan	[9]
6	42	M	*Roussoella neopustulans*	Keratitis	Trauma by a wood fragment	Natamycin	Brazil	our study

Note: Clinical cases published of *Roussouella* genus infection reported in literature since 2014 to 2022 were reviewed. The cases were searched in PubMed using search terms “*Roussouella*”. ITCZ, Itraconazole; M, Male; MCFG, Micafungi; N, Number; PSCZ, Posaconazole; VRCZ, Voriconazole.

## Data Availability

The data presented in this study are openly available in NCBI, accession number OL799163 and OL799109.

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
