# Peer review of "An Atypical Etiology of Fungal Keratitis Caused by Roussoella neopustulans"

_jof, 2022, doi:10.3390/jof8050507_

Round 1

Reviewer 1 Report

1) The current case report of the Roussoella neopustulans causing fungal keratitis is interesting.  What are the other complications caused by this pathogen?

2) Authors should provide the details of the amplification of the internal transcribed spacer rather than mentioning the use of standard manufacturer's instructions. 

3) Authors should perform the antifungal susceptibility testing on the isolate and provide information regarding its susceptibility or resistance to available antifungal agents.

Author Response

Response to Reviewer 1 Comments

Point 1: The current case report of the Roussoella neopustulans causing fungal keratitis is interesting.  What are the other complications caused by this pathogen?

Response 1: We would like to thank you for the review and comments. Roussoella neopustulans are firstly characterizes as saprobic on decaying bamboo culm. This is the first case of infection associated to R. neopustulans, for this reason no other complications were described in the literature. According your suggestions, a table with clinical cases reported in the Roussoella genus was provided in the discussion.

Point 2: Authors should provide the details of the amplification of the internal transcribed spacer rather than mentioning the use of standard manufacturer's instructions.

Response 2: According your suggestions the details of the amplification and sequencing was provided. The sentence was included “PCR reaction mixtures were made with a total volume of 12.5 μL, (1× PCR buffer, 2.0 mM MgCl2, 25 μM deoxynucleoside triphosphates (dNTPs), 0.5 μM of each forward and reverse primer, 1 U of Taq DNA polymerase (Ludwig Biotec, Bela Vista, Brazil), and 10 ng of genomic DNA. PCR parameters for amplification were 95 °C for 4 min, followed by 35 cycles consisting of 95 °C for 45 s, 52 °C for 30 s, and 72 °C for 2 min, and a delay at 72 °C for 7 min, performed in an ABI Prism 2720 thermocycler (Applied Biosystems, Foster City, USA). Amplicons were sequenced with BigDye Terminator cycle sequencing kit v. 3.1(Applied Biosystems, Foster City, CA, USA) according to the manufacturer's instructions, using the same primers of the PCR, and the amplification condition as follow: 95 °C for 1 min, followed by 30 cycles consisting of 95 °C for 10 s, 50 °C for 5 s and 60 °C.”

Point 3: Authors should perform the antifungal susceptibility testing on the isolate and provide information regarding its susceptibility or resistance to available antifungal agents.

Response 2: We are grateful for your suggesting; however, the antifungal susceptibility testing is not standardized methodology in our institution which could lead non-consistent data.

Sincerely,

The Authors

Reviewer 2 Report

The manuscript is well written in an engaging and lively style.
The level is appropriate to the readership of JoF. I have some comments.
I would strongly advise the author to rewrite their introduction to produce a more contextualized introduction to Roussoella neopustulans. 
Case reports
Write the report according to the CARE guide.
Describe the primers used for sequencing.

In the “Discussion” section I would have wished to see more information, for example, a table with reported cases of Roussoella neopustulans. 

Author Response

Response to Reviewer 2 Comments

Point 1: The manuscript is well written in an engaging and lively style.

The level is appropriate to the readership of JoF. I have some comments.

I would strongly advise the author to rewrite their introduction to produce a more contextualized introduction to Roussoella neopustulans.

Response 1: We would like to thank you for the review and comments. This is the first case of infection associated to Roussoella neopustulans. According your suggestions, the sentence was included in the introduction section “Corneal smears and culture, the current gold standard mode of diagnosis, demonstrate an increasing trend in the fungal keratitis and a wide variety of fungal pathogens (>100 species) [1] including new and rare species. Mochizuki et al., (2017) [9] related the first case of Roussoella solani as etiologic agent involved in keratomycosis in a human host from Japan. Species belong to Roussoella genus are manly described as saprobic, e.g. Roussoella nitidula, Roussoella intermedia, Roussoella japanensis and Roussoella neopustulans isolated from decaying bamboo culm [10]. Few cases reported subcutaneous mycoses involved Roussoella percutanea Ahmed et al. (2014) [11, 12; 13, 14]..”

Point 2: Write the report according to the CARE guide.

Response 2: According your suggestion the section “2. Case Presentation” was re-write according to the CARE guide.

Point 3: Describe the primers used for sequencing.

Response 2:   The description of sequence primers used for sequencing was included by the sentence “Amplicons were sequenced with BigDye Terminator cycle sequencing kit v. 3.1(Applied Biosystems, Foster City, CA, USA) according to the manufacturer's instructions, using the same primers of the PCR,”.

Point 3: In the “Discussion” section I would have wished to see more information, for example, a table with reported cases of Roussoella neopustulans.

Response 2: Few relates in associated with Roussoella genus are available in literature, and this is the first report of infection by Roussoella neopustulans. However, a table was added in the discussion with a description of clinical infections by Roussoella genus.

Sincerely,

The Authors